# Progressive Skeletonization: Trimming more fat from a network at initialization

**Pau de Jorge**[*]
University of Oxford &
NAVER LABS Europe[†]

**Amartya Sanyal**
University of Oxford &
The Alan Turing Institute, London, UK

**Harkirat S. Behl**
University of Oxford

**Philip H. S. Torr**
University of Oxford

**Grégory Rogez**
NAVER LABS Europe

**Puneet K. Dokania**
University of Oxford &
Five AI Limited

## Abstract

Recent studies have shown that skeletonization (pruning parameters) of networks *at initialization* provides all the practical benefits of sparsity both at inference and training time, while only marginally degrading their performance. However, we observe that beyond a certain level of sparsity (approx 95%), these approaches fail to preserve the network performance, and to our surprise, in many cases perform even worse than trivial random pruning. To this end, we propose an objective to find a skeletonized network with maximum *foresight connection sensitivity* (FORCE) whereby the trainability, in terms of connection sensitivity, of a pruned network is taken into consideration. We then propose two approximate procedures to maximize our objective (1) Iterative SNIP: allows parameters that were unimportant at earlier stages of skeletonization to become important at later stages; and (2) FORCE: iterative process that allows exploration by allowing already pruned parameters to resurrect at later stages of skeletonization. Empirical analysis on a large suite of experiments show that our approach, while providing at least as good a performance as other recent approaches on moderate pruning levels, provide remarkably improved performance on higher pruning levels (could remove up to 99.5% parameters while keeping the networks trainable).

## 1 Introduction

The majority of pruning algorithms for Deep Neural Networks require training dense models and often fine-tuning sparse sub-networks in order to obtain their pruned counterparts. In Frankle & Carbin (2019), the authors provide empirical evidence to support the hypothesis that there exist sparse sub-networks that can be trained from scratch to achieve similar performance as the dense ones. However, their method to find such sub-networks requires training the full-sized model and intermediate sub-networks, making the process much more expensive.

Recently, Lee et al. (2019) presented SNIP. Building upon almost a three decades old saliency criterion for pruning trained models (Mozer & Smolensky, 1989), they are able to predict, at initialization, the importance each weight will have later in training. Pruning at initialization methods are much cheaper than conventional pruning methods. Moreover, while traditional pruning methods can help accelerate inference tasks, pruning at initialization may go one step further and provide the same benefits at train time Elsen et al. (2020).

Wang et al. (2020) (GRASP) noted that after applying the pruning mask, gradients are modified due to non-trivial interactions between weights. Thus, maximizing SNIP criterion before pruning might be sub-optimal. They present an approximation to maximize the gradient norm after pruning, where they treat pruning as a perturbation on the weight matrix and use the first order Taylor's approximation. While they show improved performance, their approximation involves computing a Hessian-vector product which is expensive both in terms of memory and computation.

---

[*]Correspondence to `pau@robots.ox.ac.uk`
[†]`www.europe.naverlabs.com`

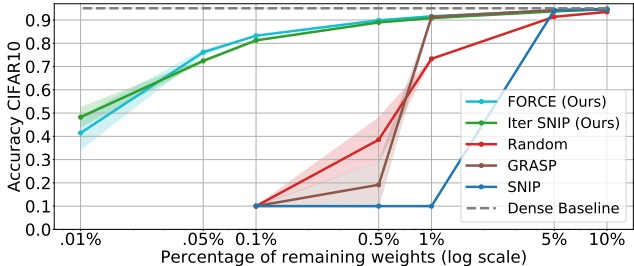

Figure 1: Test accuracies on CIFAR-10 (ResNet50) for different pruning methods. Each point is the average over 3 runs of prune-train-test. The shaded areas denote the standard deviation of the runs (too small to be visible in some cases). Random corresponds to removing connections uniformly.

We argue that both SNIP and GRASP approximations of the gradients after pruning do not hold for high pruning levels, where a large portion of the weights are removed at once. In this work, while we rely on the saliency criteria introduced by Mozer & Smolensky (1989), we optimize what this saliency would be *after* pruning, rather than *before*. Hence, we name our criteria *Foresight Connection sEnsitivity* (FORCE). We introduce two approximate procedures to progressively optimize our objective. The first, which turns out to be equivalent to applying SNIP iteratively, removes a small fraction of weights at each step and re-computes the gradients after each pruning round. This allows to take into account the intricate interactions between weights, re-adjusting the importance of connections at each step. The second procedure, which we name FORCE, is also iterative in nature, but contrary to the first, it allows pruned parameters to resurrect. Hence, it supports exploration, which otherwise is not possible in the case of iterative SNIP. Moreover, one-shot SNIP can be viewed as a particular case of using only one iteration. Empirically, we find that both SNIP and GRASP have a sharp drop in performance when targeting higher pruning levels. Surprisingly, they perform even worse than random pruning as can be seen in Fig 1. In contrast, our proposed pruning procedures prove to be significantly more robust on a wide range of pruning levels.

## 2 RELATED WORK

**Pruning trained models** Most of the pruning works follow the *train – prune – fine-tune* cycle (Mozer & Smolensky, 1989; LeCun et al., 1990; Hassibi et al., 1993; Han et al., 2015; Molchanov et al., 2017; Guo et al., 2016), which requires training the dense network until convergence, followed by multiple iterations of pruning and fine-tuning until a target sparsity is reached. Particularly, Molchanov et al. (2017) present a criterion very similar to Mozer & Smolensky (1989) and therefore similar to Lee et al. (2019) and our FORCE, but they focus on pruning whole neurons, and involve training rounds while pruning. Frankle & Carbin (2019) and Frankle et al. (2020) showed that it was possible to find sparse sub-networks that, when trained from scratch or an early training iteration, were able to match or even surpass the performance of their dense counterparts. Nevertheless, to find them they use a costly procedure based on Han et al. (2015). All these methods rely on having a trained network, thus, they are not applicable before training. In contrast, our algorithm is able to find a trainable sub-network with randomly initialized weights. Making the overall pruning cost much cheaper and presenting an opportunity to leverage the sparsity during training as well.

**Induce sparsity during training** Another popular approach has been to induce sparsity during training. This can be achieved by modifying the loss function to consider sparsity as part of the optimization (Chauvin, 1989; Carreira-Perpiñán & Idelbayev, 2018; Louizos et al., 2018) or by dynamically pruning during training (Bellec et al., 2018; Mocanu et al., 2018; Mostafa & Wang, 2019; Dai et al., 2019; Dettmers & Zettlemoyer, 2020; Lin et al., 2020; Kusupati et al., 2020; Evci et al., 2019). These methods are usually cheaper than pruning after training, but they still need to train the network to select the final sparse sub-network. We focus on finding sparse sub-networks before any weight update, which is not directly comparable.

**Pruning at initialization** These methods present a significant leap with respect to other pruning methods. While traditional pruning mechanisms focused on bringing speed-up and memory reduction at inference time, pruning at initialization methods bring the same gains both at training and inference time. Moreover, they can be seen as a form of Neural Architecture Search (Zoph & Le, 2016) to find more *efficient* network topologies. Thus, they have both a theoretical and practical interest.

Lee et al. (2019) presented SNIP, a method to estimate, at initialization, the importance that each weight could have later during training. SNIP analyses the effect of each weight on the loss function when perturbed at initialization. In Lee et al. (2020), the authors studied pruning at initialization from a signal propagation perspective, focusing on the initialization scheme. Recently, Wang et al. (2020) proposed GRASP, a different method based on the gradient norm after pruning and showed a significant improvement for higher levels of sparsity. However, neither SNIP nor GRASP perform sufficiently well when larger compressions and speed-ups are required and a larger fraction of the weights need to be pruned. In this paper, we analyse the approximations made by SNIP and GRASP, and present a more suitable solution to maximize the saliency after pruning.

## 3 PROBLEM FORMULATION: PRUNING AT INITIALIZATION

Given a dataset $\mathcal{D} = \{(\mathbf{x}_i, \mathbf{y}_i)\}_{i=1}^n$, the training of a neural network $f$ parameterized by $\boldsymbol{\theta} \in \mathbb{R}^m$ can be written as minimizing the following empirical risk:

$$\underset{\boldsymbol{\theta}}{\arg\min} \ \frac{1}{n} \sum_i \mathcal{L}((f(\mathbf{x}_i; \boldsymbol{\theta})), \mathbf{y}_i) \quad \text{s.t. } \boldsymbol{\theta} \in \mathcal{C}, \tag{1}$$

where $\mathcal{L}$ and $\mathcal{C}$ denote the loss function and the constraint set, respectively. Unconstrained (standard) training corresponds to $\mathcal{C} = \mathbb{R}^m$. Assuming we have access to the gradients (batch-wise) of the empirical risk, an optimization algorithm (*e.g.* SGD) is generally used to optimize the above objective, that, during the optimization process, produces a sequence of iterates $\{\boldsymbol{\theta}_i\}_{i=0}^T$, where $\boldsymbol{\theta}_0$ and $\boldsymbol{\theta}_T$ denote the initial and the final (optimal) parameters, respectively. Given a target sparsity level of $k < m$, the *general* parameter pruning problem involves $\mathcal{C}$ with a constraint $\|\boldsymbol{\theta}_T\|_0 \leq k$, *i.e.*, the final optimal iterate must have a maximum of $k$ non-zero elements. Note that there is no such constraint with the intermediate iterates.

Pruning at initialization, the main focus of this work, adds further restrictions to the above mentioned formulation by constraining all the iterates to lie in a *fixed* subspace of $\mathcal{C}$. Precisely, the constraints are to find an initialization $\boldsymbol{\theta}_0$ such that $\|\boldsymbol{\theta}_0\|_0 \leq k$ [1], and the intermediate iterates are $\boldsymbol{\theta}_i \in \bar{\mathcal{C}} \subset \mathcal{C}$, $\forall i \in \{1, \ldots, T\}$, where $\bar{\mathcal{C}}$ is the subspace of $\mathbb{R}^m$ spanned by the natural basis vectors $\{\mathbf{e}_j\}_{j \in \text{supp}(\boldsymbol{\theta}_0)}$. Here, $\text{supp}(\boldsymbol{\theta}_0)$ denotes the support of $\boldsymbol{\theta}_0$, *i.e.*, the set of indices with non-zero entries. The first condition defines the sub-network at initialization with $k$ parameters, and the second fixes its topology throughout the training process. Since there are $\binom{m}{k}$ such possible sub-spaces, exhaustive search to find the optimal sub-space to optimize (1) is impractical as it would require training $\binom{m}{k}$ neural networks. Below we discuss two recent approaches that circumvent this problem by maximizing a hand-designed data-dependent objective function. These objectives are tailored to preserve some relationships between the parameters, the loss, and the dataset, that *might* be sufficient to obtain a reliable $\boldsymbol{\theta}_0$. For the ease of notation, we will use $\boldsymbol{\theta}$ to denote the dense initialization.

**SNIP** Lee et al. (2019) present a method based on the saliency criterion from Mozer & Smolensky (1989). They add a key insight and show this criteria works surprisingly well to predict, at initialization, the importance each connection will have during training. The idea is to preserve the parameters that will have maximum impact on the loss when perturbed. Let $\boldsymbol{c} \in \{0, 1\}^m$ be a binary vector, and $\odot$ the Hadamard product. Then, the *connection sensitivity* in SNIP is computed as:

$$\boldsymbol{g}(\boldsymbol{\theta}) := \left. \frac{\partial \mathcal{L}(\boldsymbol{\theta} \odot \boldsymbol{c})}{\partial \boldsymbol{c}} \right|_{\boldsymbol{c}=1} = \frac{\partial \mathcal{L}(\boldsymbol{\theta})}{\partial \boldsymbol{\theta}} \odot \boldsymbol{\theta}. \tag{2}$$

Once $\boldsymbol{g}(\boldsymbol{\theta})$ is obtained, the parameters corresponding to the top-$k$ values of $|\boldsymbol{g}(\boldsymbol{\theta})_i|$ are then kept. Intuitively, SNIP favors those weights that are far from the origin *and* provide high gradients (irrespective of the direction). We note that SNIP objective can be written as the following problem:

$$\max_{\boldsymbol{c}} S(\boldsymbol{\theta}, \boldsymbol{c}) := \sum_{i \in \text{supp}(\boldsymbol{c})} |\theta_i \nabla \mathcal{L}(\boldsymbol{\theta})_i| \quad \text{s.t. } \boldsymbol{c} \in \{0, 1\}^m, \ \|\boldsymbol{c}\|_0 = k. \tag{3}$$

It is trivial to note that the optimal solution to the above problem can be obtained by selecting the indices corresponding to the top-$k$ values of $|\theta_i \nabla \mathcal{L}(\boldsymbol{\theta})_i|$.

---

[1] In practice, as will be done in this work as well, a subset of a *given* dense initialization is found using some saliency criterion (will be discussed soon), however, note that our problem statement is more general than that.

**GRASP** Wang et al. (2020) note that the SNIP saliency is measuring the *connection sensitivity* of the weights *before* pruning, however, it is likely to change after pruning. Moreover, they argue that, at initialization, it is more important to preserve the gradient signal than the loss itself. They propose to use as saliency the gradient norm of the loss $\Delta\mathcal{L}(\boldsymbol{\theta}) = \nabla\mathcal{L}(\boldsymbol{\theta})^T\nabla\mathcal{L}(\boldsymbol{\theta})$, but measured *after* pruning. To maximize it, Wang et al. (2020) adopt the same approximation introduced in LeCun et al. (1990) and treat pruning as a perturbation on the initial weights. Their method is equivalent to solving:

$$\max_{\boldsymbol{c}} G(\boldsymbol{\theta}, \boldsymbol{c}) := \sum_{\{i:\ c_i=0\}} -\theta_i\,[\mathbf{Hg}]_i \quad \text{s.t. } \boldsymbol{c} \in \{0,1\}^m, \ \|\boldsymbol{c}\|_0 = k. \tag{4}$$

Where $\mathbf{H}$ and $\mathbf{g}$ denote the Hessian and the gradient of the loss respectively.

## 4 FORESIGHT CONNECTION SENSITIVITY

Since removing connections of a neural network will have significant impact on its forward and backward signals, we are interested in obtaining a pruned network that is easy to train. We use *connection sensitivity* of the loss function as a proxy for the so-called trainability of a network. To this end, we first define connection sensitivity *after* pruning which we name *Foresight Connection sEnsitivity* (FORCE), and then propose two procedures to optimize it in order to obtain the desired pruned network. Let $\bar{\boldsymbol{\theta}} = \boldsymbol{\theta} \odot \boldsymbol{c}$ denotes the pruned parameters once a binary mask $\boldsymbol{c}$ with $\|\boldsymbol{c}\|_0 = k \leq m$ is applied. The FORCE at $\bar{\boldsymbol{\theta}}$ for a given mask $\hat{\boldsymbol{c}}$ is then obtained as:

$$\boldsymbol{g}(\bar{\boldsymbol{\theta}}) := \left.\frac{\partial\mathcal{L}(\bar{\boldsymbol{\theta}})}{\partial\boldsymbol{c}}\right|_{\boldsymbol{c}=\hat{\boldsymbol{c}}} = \left.\frac{\partial\mathcal{L}(\bar{\boldsymbol{\theta}})}{\partial\bar{\boldsymbol{\theta}}}\right|_{\boldsymbol{c}=\hat{\boldsymbol{c}}} \odot \left.\frac{\partial\bar{\boldsymbol{\theta}}}{\partial\boldsymbol{c}}\right|_{\boldsymbol{c}=\hat{\boldsymbol{c}}} = \left.\frac{\partial\mathcal{L}(\bar{\boldsymbol{\theta}})}{\partial\bar{\boldsymbol{\theta}}}\right|_{\boldsymbol{c}=\hat{\boldsymbol{c}}} \odot \boldsymbol{\theta}. \tag{5}$$

The last equality is obtained by rewriting $\bar{\boldsymbol{\theta}}$ as $\text{diag}(\boldsymbol{\theta})\boldsymbol{c}$, where $\text{diag}(\boldsymbol{\theta})$ is a diagonal matrix with $\boldsymbol{\theta}$ as its elements, and then differentiating w.r.t. $\boldsymbol{c}$. Note, when $k < m$, the sub-network $\bar{\boldsymbol{\theta}}$ is obtained by *removing* connections corresponding to all the weights for which the binary variable is zero. Therefore, only the weights corresponding to the indices for which $\boldsymbol{c}(i) = 1$ contribute in equation (5), all other weights do not participate in forward and backward propagation and are to be ignored. We now discuss the crucial differences between our formulation (5), SNIP (2) and GRASP (4).

- When $\hat{\boldsymbol{c}} = \mathbf{1}$, the formulation is exactly the same as the connection sensitivity used in SNIP. However, $\hat{\boldsymbol{c}} = \mathbf{1}$ is too restrictive in the sense that it assumes that all the parameters are active in the network and they are removed one by one *with replacement*, therefore, it fails to capture the impact of removing a group of parameters.

- Our formulation uses weights and gradients corresponding to $\bar{\boldsymbol{\theta}}$ thus, compared to SNIP, provides a better indication of the training dynamics of the pruned network. However, GRASP formulation is based on the assumption that pruning is a small perturbation on the Gradient Norm which, also shown experimentally, is not always a reliable assumption.

- When $\|\hat{\boldsymbol{c}}\|_0 \ll \|\mathbf{1}\|_0$, *i.e.*, extreme pruning, the gradients before and after pruning will have very different values as $\|\boldsymbol{\theta}\odot\hat{\boldsymbol{c}}\|_2 \ll \|\boldsymbol{\theta}\|_2$, making SNIP and GRASP unreliable (empirically we find SNIP and GRASP fail in the case of high sparsity).

**FORCE saliency** Note FORCE (5) is defined for a given sub-network which is unknown *a priori*, as our objective itself is to find the sub-network with maximum connection sensitivity. Similar to the reformulation of SNIP in (3), the objective to find such sub-network corresponding to the foresight connection sensitivity can be written as:

$$\max_{\boldsymbol{c}} S(\boldsymbol{\theta}, \boldsymbol{c}) := \sum_{i\in\text{supp}(\boldsymbol{c})} |\theta_i\,\nabla\mathcal{L}(\boldsymbol{\theta}\odot\boldsymbol{c})_i| \quad \text{s.t. } \boldsymbol{c} \in \{0,1\}^m, \ \|\boldsymbol{c}\|_0 = k. \tag{6}$$

Here $\nabla\mathcal{L}(\boldsymbol{\theta}\odot\boldsymbol{c})_i$ represents the $i$-th index of $\left.\frac{\partial\mathcal{L}(\bar{\boldsymbol{\theta}})}{\partial\bar{\boldsymbol{\theta}}}\right|_{\boldsymbol{c}}$. As opposed to (3), finding the optimal solution of (6) is non trivial as it requires computing the gradients of all possible $\binom{m}{k}$ sub-networks in order to find the one with maximum sensitivity. To this end, we present two approximate solutions to the above problem that primarily involve (i) progressively increasing the degree of pruning, and (ii) solving an approximation of (6) at each stage of pruning.

**Progressive Pruning (Iterative SNIP)** Let $k$ be the number of parameters to be kept after pruning. Let us assume that we know a schedule (will be discussed later) to divide $k$ into a set of natural

numbers $\{k_t\}_{t=1}^{T}$ such that $k_t > k_{t+1}$ and $k_T = k$. Now, given the mask $\boldsymbol{c}_t$ corresponding to $k_t$, pruning from $k_t$ to $k_{t+1}$ can be formulated using the connection sensitivity (5) as:

$$\boldsymbol{c}_{t+1} = \arg\max_{\boldsymbol{c}} S(\bar{\boldsymbol{\theta}}, \boldsymbol{c}) \quad \text{s.t.} \ \ \boldsymbol{c} \in \{0,1\}^m, \ \ \|\boldsymbol{c}\|_0 = k_{t+1}, \ \boldsymbol{c} \odot \boldsymbol{c}_t = \boldsymbol{c}, \tag{7}$$

where $\bar{\boldsymbol{\theta}} = \boldsymbol{\theta} \odot \boldsymbol{c}_t$. The additional constraint $\boldsymbol{c} \odot \boldsymbol{c}_t = \boldsymbol{c}$ ensures that no parameter that had been pruned earlier is activated again. Assuming that the pruning schedule ensures a smooth transition from one topology to another ($\|\boldsymbol{c}_t\|_0 \approx \|\boldsymbol{c}_{t+1}\|_0$) such that the *gradient approximation* $\left.\frac{\partial \mathcal{L}(\bar{\boldsymbol{\theta}})}{\partial \boldsymbol{\theta}}\right|_{\boldsymbol{c}_t} \approx \left.\frac{\partial \mathcal{L}(\bar{\boldsymbol{\theta}})}{\partial \boldsymbol{\theta}}\right|_{\boldsymbol{c}_{t+1}}$ is valid, (7) can be approximated as solving (3) at $\bar{\boldsymbol{\theta}}$. Thus, for a given schedule over $k$, our first approximate solution to (6) involves solving (3) iteratively. This allows re-assessing the importance of connections after changing the sub-network. For a schedule with $T = 1$, we recover SNIP where a crude *gradient approximation* between the dense network $\boldsymbol{c}_0 = \mathbf{1}$ and the final mask $\boldsymbol{c}$ is being used. This approach of ours turns out to be algorithmically similar to a concurrent work (Verdenius et al., 2020). However, our motivation comes from a novel objective function (5) which also gives place to our second approach (FORCE). Tanaka et al. (2020) also concurrently study the effect of iterative pruning and report, similar to our findings, pruning progressively is needed for high sparsities.

**Progressive Sparsification (FORCE)** The constraint $\boldsymbol{c} \odot \boldsymbol{c}_t = \boldsymbol{c}$ in (7) (Iterative SNIP) might be restrictive in the sense that while re-assessing the importance of *unpruned* parameters, it does not allow previously *pruned* parameters to resurrect (even if they could become important). This hinders exploration which can be unfavourable in finding a suitable sub-network. Here we remove this constraint, meaning, the weights for which $\boldsymbol{c}(i) = 0$ are *not* removed from the network, rather they are assigned a value of zero. Therefore, while not contributing to the forward signal, they might have a non-zero gradient. This relaxation modifies the saliency in (5) whereby the gradient is now computed at a *sparsified* network instead of a *pruned* network. Similar to the above approach, we sparsify the network progressively and once the desired sparsity is reached, all connections with $\boldsymbol{c}(i) = 0$ are *pruned*. Note, the step of removing zero weights is valid if removing such connections does not adversely impact the gradient flow of the unpruned parameters. We, in fact, found it to be true in our experiments shown in Fig 7 (Appendix). However, this assumption might not hold always.

An overview of Iterative SNIP and FORCE is presented in Algorithm 1.

**Sparsity schedule** Both the above discussed iterative procedures approximately optimize (5), however, they depend on a sparsity/pruning schedule favouring *small* steps to be able to reliably apply the mentioned *gradient approximation*. One such valid schedule would be where the portion of newly removed weights with respect to the remaining weights is small. We find a simple exponential decay schedule, defined below, to work very well on all our experiments:

$$\text{Exp mode:} \ \ k_t = \exp\left\{\alpha \log k + (1 - \alpha) \log m\right\}, \ \ \alpha = \frac{t}{T}. \tag{8}$$

In section 5.3 we empirically show that these methods are very robust to the hyperparameter $T$.

**Some theoretical insights** When pruning weights gradually, we are looking for the best possible sub-network in a neighbourhood defined by the previous mask and the amount of weights removed at that step. The problem being non-convex and non-smooth makes it challenging to prove if the mask

---

**Algorithm 1** FORCE/Iter SNIP algorithms to find a pruning mask

1: **Inputs:** Training set $\mathcal{D}$, final sparsity $k$, number of steps $T$, weights $\boldsymbol{\theta}_0 \in \mathbb{R}^m$.
2: Obtain $\{k_t\}_{t=1:T}$ using the chosen schedule (refer to Eq (8))
3: Define intial mask $\boldsymbol{c_0} = \mathbf{1}$
4: **for** $t = 0, \ldots, T - 1$ **do**
5:     Sample mini-batch $\{z_i\}_{i=1}^{n}$ from $\mathcal{D}$
6:     Define $\bar{\boldsymbol{\theta}} = \boldsymbol{\theta} \odot \boldsymbol{c}_t$ (as *sparsified* (FORCE) vs *pruned* (Iter SNIP) network)
7:     Compute $\boldsymbol{g}(\bar{\boldsymbol{\theta}})$ (refer to Eq (5) )
8:     $I = \{i_1, \ldots, i_{k_{t+1}}\}$ are top-$k_{t+1}$ values of $|\boldsymbol{g}_i|$
9:     Build $\boldsymbol{c}_{t+1}$ by setting to 0 all indices not included in $I$.
10: **end for**
11: **Return:** $\boldsymbol{c}_T$.

---

obtained by our method is globally optimal. However, in Appendix D we prove that each intermediate mask obtained with Iterative SNIP is indeed an approximate local minima, where the degree of sub-optimality increases with the pruning step size. This gives some validation on why SNIP fails on higher sparsity. We can not provide the same guarantees for FORCE (there is no obvious link between the step size and the distance between masks), nevertheless, we empirically observe that FORCE is quite robust and more often than not improves over the performance of Iterative SNIP, which is not able to recover weights once pruned. We present further analysis in Appendix C.4.

## 5 EXPERIMENTS

In the following we evaluate the efficacy of our approaches accross different architectures and datasets. Training settings, architecture descriptions, and implementation details are provided in Appendix A.

### 5.1 RESULTS ON CIFAR-10

Fig 2 compares the accuracy of the described iterative approaches with both SNIP and GRASP. We also report the performance of a dense and a random pruning baseline. Both SNIP and GRASP consider a single batch to approximate the saliencies, while we employ a different batch of data at each stage of our gradual skeletonization process. For a fair comparison, and to understand how the number of batches impacts performance, we also run these methods averaging the saliencies over $T$ batches, where $T$ is the number of iterations. SNIP-MB and GRASP-MB respectively refer to these multi-batch (MB) counterparts. In these experiments, we use $T = 300$. We study the hyper-parameter robustness regarding $T$ later in section 5.3.

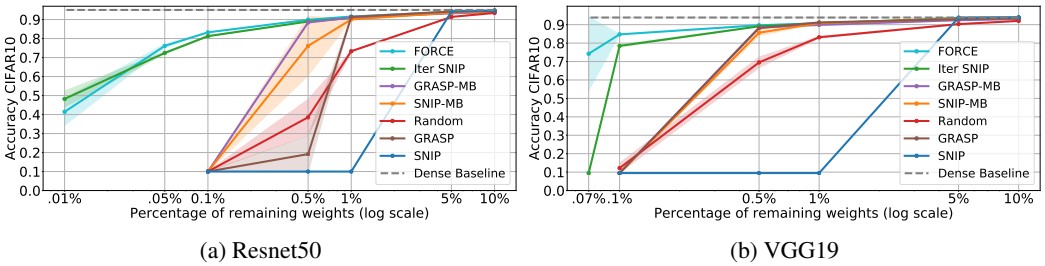

(a) Resnet50          (b) VGG19

Figure 2: Test accuracies on CIFAR-10 for different pruning methods. With increased number of batches (-MB) one-shot methods are more robust at higher sparsity levels, but our gradual pruning approaches can go even further. Moreover, FORCE consistently reaches higher accuracy than other methods across most sparsity levels. Each point is an average of $\geq 3$ runs of prune-train-test. The shaded areas denote the standard deviation of the runs (too small to be visible in some cases). Note that in (b), GRASP and GRASP-MB overlap.

We observe that for moderate sparsity levels, one batch is sufficient for both SNIP and GRASP as reported in Lee et al. (2019); Wang et al. (2020). However, as we increase the level of sparsity, the performance of SNIP and GRASP degrades dramatically. For example, at $99.0\%$ sparsity, SNIP drops down to $10\%$ accuracy for both ResNet50 and VGG19, which is equivalent to random guessing as there are 10 classes. Note, in the case of randomly pruned networks, accuracy is nearly $75\%$ and $82\%$ for ResNet50 and VGG19, respectively, which is significantly better than the performance of SNIP. However, to our surprise, just using multiple batches to compute the connection sensitivity used in SNIP improves it from $10\%$ to almost $90\%$. This clearly indicates that a better approximation of the connection sensitivity is necessary for good performance in the case of high sparsity regime. Similar trends, although not this extreme, can be observed in the case of GRASP as well. On the other hand, gradual pruning approaches are much more robust in terms of sparsity for example, in the case of $99.9\%$ pruning, while one-shot approaches perform as good as a random classifier (nearly $10\%$ accuracy), both FORCE and Iterative SNIP obtain more than $80\%$ accuracy. While the accuracies obtained at higher sparsities might have degraded too much for some use cases, we argue this is an encouraging result, as no approach before has pruned a network at initialization to such extremes while keeping the network trainable and these results might encourage the community to improve the performance further. Finally, gradual pruning methods consistently improve over other methods even at moderate sparsity levels (refer to Fig 5), this motivates the use of FORCE or Iterative SNIP instead of other methods by default at any sparsity regime. Moreover, the additional cost of using iterative

pruning instead of SNIP-MB is negligible compared to the cost of training and is significantly cheaper than GRASP-MB, further discussed in section 5.3.

(a) CIFAR 100 - Resnet50

(b) CIFAR 100 - VGG19

(c) Tiny Imagenet - Resnet50

(d) Tiny Imagenet - VGG19

Figure 3: Test accuracies on CIFAR-100 and Tiny Imagenet for different pruning methods. Each point is the average over 3 runs of prune-train-test. The shaded areas denote the standard deviation of the runs (too small to be visible in some cases).

## 5.2 RESULTS ON LARGER DATASETS

We now present experiments on large datasets. Wang et al. (2020) and Lee et al. (2019) suggest using a batch of size ~10 times the number of classes, which is very large in these experiments. Instead, for memory efficiency, we average the saliencies over several mini-batches. For CIFAR100 and Tiny-ImageNet, we average 10 and 20 batches per iteration respectively, with 128 examples per batch. As we increase the number of batches per iteration, computing the pruning mask becomes more expensive. From Fig 4, we observe that the accuracy converges after just a few iterations. Thus, for the following experiments we used 60 iterations. For a fair comparison, we run SNIP and GRASP with $T \times B$ batches, where $T$ is the number of iterations and $B$ the number of batches per iteration in our method. We find the results, presented in Fig 3, consistent with trends in CIFAR-10.

In the case of Imagenet, we use a batch size of 256 examples and 40 batches per iteration. We use the official implementation of VGG19 with batch norm and Resnet50 from Paszke et al. (2017). As presented in Table 1, gradual pruning methods are consistently better than SNIP, with a larger gap as we increase sparsity. We would like to emphasize that FORCE is able to prune 90% of the weights of VGG while losing less than 3% of the accuracy, we find this remarkable for a method that prunes before any weight update. Interestingly, GRASP performs better than other methods at 95% sparsity (VGG), moreover, it also slightly surpasses FORCE for Resnet50 at 90%, however, it under-performs random pruning at 95%. In fact, we find all other methods to perform worse than random pruning for Resnet50. We hypothesize that, for a much more challenging task (Imagenet with 1000 classes), Resnet50 architecture might not be extremely overparametrized. For instance,

Table 1: Test accuracies on Imagenet for different pruning methods and sparsities.

| Network | VGG19 | | | | Resnet50 | | | |
|---|---|---|---|---|---|---|---|---|
| **Sparsity percentage** | 90% | | 95% | | 90% | | 95% | |
| Accuracy | Top-1 | Top-5 | Top-1 | Top-5 | Top-1 | Top-5 | Top-1 | Top-5 |
| (Baseline) | 73.1 | 91.3 | | | 75.6 | 92.8 | | |
| FORCE (Ours) | **70.2** | **89.5** | 65.8 | 86.8 | 64.9 | 86.5 | **59.0** | **82.3** |
| Iter SNIP (Ours) | 69.8 | 89.5 | 65.9 | 86.9 | 63.7 | 85.5 | 54.7 | 78.9 |
| GRASP Wang et al. (2020) | 69.5 | 89.2 | **67.6** | **87.8** | **65.4** | **86.7** | 46.2 | 66.0 |
| SNIP Lee et al. (2019) | 68.5 | 88.8 | 63.8 | 86.0 | 61.5 | 83.9 | 44.3 | 69.6 |
| Random | 64.2 | 86.0 | 56.6 | 81.0 | 64.6 | 86.0 | 57.2 | 80.8 |

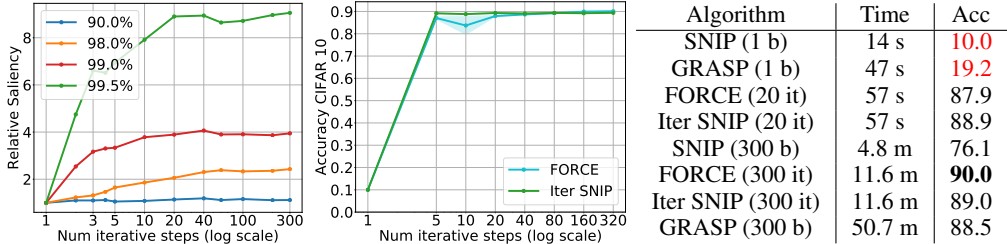

| Algorithm | Time | Acc |
|---|---|---|
| SNIP (1 b) | 14 s | 10.0 |
| GRASP (1 b) | 47 s | 19.2 |
| FORCE (20 it) | 57 s | 87.9 |
| Iter SNIP (20 it) | 57 s | 88.9 |
| SNIP (300 b) | 4.8 m | 76.1 |
| FORCE (300 it) | 11.6 m | **90.0** |
| Iter SNIP (300 it) | 11.6 m | 89.0 |
| GRASP (300 b) | 50.7 m | 88.5 |

Figure 4: **Left:** FORCE saliency (6) obtained with (7) when varying $T$ normalized by the saliency with one-shot SNIP ($T = 1$). Pruning iteratively brings more gains for higher sparsity levels. Error bars not shown for better visualization. **Middle:** Test acc pruning with FORCE and Iter SNIP at 99.5% sparsity for different $T$. Both methods are extremely robust to the choice of $T$. **Right:** Wall time to compute pruning masks for CIFAR10/Resnet50/TeslaK40m vs acc at 99.5% sparsity; ($x$ b) means we used $x$ batches to compute the gradients while ($x$ it) denotes we used $x$ pruning iterations, with one batch per iteration. Numbers in red indicate performance below random pruning.

VGG19 has 143.68M parameters while Resnet50 uses 25.56M (refer to Table 2). On the other hand, the fact that random pruning can yield relatively trainable architectures for these sparsity levels is somewhat surprising and might indicate that there still is room for improvement in this direction. Results seem to indicate that the FORCE saliency is a step in the right direction and we hypothesize further improvements on its optimization might lead to even better performance. In Appendix C.6, we show superior performance of our approach on the **Mobilenet-v2 architecture** (Sandler et al., 2018) as well, which is much more "slim" than Resnet and VGG [2].

## 5.3 ANALYSIS

**Saliency optimization** To experimentally validate our approach (7), we conduct an ablation study where we compute the FORCE saliency after pruning (5) while varying the number of iterations $T$ for different sparsity levels. In Fig 4 (left) we present the relative change in saliency as we vary the number of iterations $T$, note when $T = 1$ we recover one-shot SNIP. As expected, for moderate levels of sparsity, using multiple iterations does not have a significant impact on the saliency. Nevertheless, as we target higher sparsity levels, we can see that the saliency can be better optimized when pruning iteratively. In Appendix C.1 we include results for FORCE where we observe similar trends.

**Hyperparameter robustness** As shown in Figures 2 and 3, for low sparsity levels, all methods are comparable, but as we move to higher sparsity levels, the gap becomes larger. In Fig 4 (middle) we fix sparsity at 99.5% and study the accuracy as we vary the number of iterations $T$. Each point is averaged over 3 trials. SNIP ($T = 1$) yields sub-networks unable to train (10% acc), but as we move to iterative pruning ($T > 1$) accuracy increases up to 90% for FORCE and 89% for Iter SNIP. Moreover, accuracy is remarkably robust to the choice of $T$, the best performance for both FORCE and Iter SNIP is with more iterations, however a small number of iterations already brings a huge boost. This suggests these methods might be used by default by a user without worrying too much about hyper-parameter tuning, easily adapting the amount of iterations to their budget.

**Pruning cost** As shown in Fig 2, SNIP performance quickly degrades beyond 95% sparsity. Wang et al. (2020) suggested GRASP as a more robust alternative, however, it needs to compute a Hessian vector product which is significantly more expensive in terms of memory and time. In Fig 4 (right), we compare the time cost of different methods to obtain the pruning masks along with the corresponding accuracy. We observe that both SNIP and GRASP are fragile when using only one batch (red accuracy indicates performance below random baseline). When using multiple batches their robustness increases, but so does the pruning cost. Moreover, we find that gradual pruning based on the FORCE saliency is much cheaper than GRASP-MB when using equal amount of batches, this is because GRASP involves an expensive Hessian vector product. Thus, FORCE (or Iterative SNIP) would be preferable over GRASP-MB even when they have comparable accuracies.

**FORCE vs Iterative SNIP** Empirically, we find that FORCE tends to outperform Iter SNIP more often than not, suggesting that allowing weights to recover is indeed beneficial despite having less theoretical guarantees (see *gradient approximation* in Section 4). Thus, we would make FORCE

---

[2]Mobilenet has 2.3M params compared to 20.03M and 23.5M of VGG and Resnet, respectively.

algorithm our default choice, especially for Resnet architectures. In Appendix C.4 we empirically observe two distinct phases when pruning with FORCE. The first one involves exploration (early phase) when the amount of pruned and recovered weights seem to increase, indicating exploration of masks that are quite different from each other. The second, however, shows rapid decrease in weight recovery, indicating a phase where the algorithm converges to a more constrained topology. As opposed to Iter SNIP, the possibility of the exploration of many possible sub-networks before converging to a final topology might be the reason behind the slightly improved performance of FORCE. But this exploration comes at a price, in Fig 4 (middle and left) we observe how, despite FORCE reaching a higher accuracy when using enough steps, if we are under a highly constrained computational budget and can only afford a few pruning iterations, Iter SNIP is more likely to obtain a better pruning mask. This is indeed expected as FORCE might need more iterations to converge to a good sub-space, while Iter SNIP will be forced to converge by construction. A combination of FORCE and Iter SNIP might lead to an even better approach, we leave this for future work.

**Early pruning as an additional baseline** Our gradual pruning approaches (SNIP-MB and GRASP-MB as well) use multiple batches to obtain a pruned mask, considering that pruning can be regarded as a form of training (Mallya et al., 2018), we create another baseline for the sake of completeness. We train a network for one epoch, a similar number of iterations as used by our approach, and then use magnitude pruning to obtain the final mask, we call this approach **early pruning** (more details in Appendix C.5). Interestingly, we find that early pruning tends to perform worse than SNIP-MB (and gradual pruning) for Resnet, and shows competitive performance at low sparsity level for VGG but with a sharp drop in the performance as the sparsity level increases. Even though these experiments support the superiority of our approach, we would like to emphasize that they do not conclude that any early pruning strategy would be suboptimal compared to pruning at initialization as an effective approach in this direction might require devising a well thought objective function.

**Iterative pruning to maximize the Gradient Norm** In Sec 4, we have seen Iterative SNIP can be used to optimize the FORCE saliency. We also tried to use GRASP iteratively, however, after a few iterations the resulting networks were not trainable. Interestingly, if we apply the *gradient approximation* to GRASP saliency (instead of Taylor), we can come up with a different iterative approximation to maximize the gradient norm after pruning. We empirically observe this method is more robust than GRASP to high sparsity levels. This suggests that 1) Iterative pruning, although beneficial, can not be trivially applied to any method. 2) The gradient approximation is more general than in the context of FORCE/SNIP sensitivity. We present further details and results in Appendix E.

## 6 DISCUSSION

Pruning at initialization has become an active area of research both for its practical and theoretical interests. In this work, we discovered that existing methods mostly perform below random pruning at extreme sparsity regime. We presented FORCE, a new saliency to compute the connection sensitivity after pruning, and two approximations to progressively optimize FORCE in order to prune networks at initialization. We showed that our methods are significantly better than the existing approaches for pruning at extreme sparsity levels, and are at least as good as the existing ones for pruning at moderate sparsity levels. We also provided theoretical insights on why progressive skeletonization is beneficial at initialization, and showed that the cost of iterative methods is reasonable compared to the existing ones. Although pruning iteratively has been ubiquitous in the pruning community, it was not evident that pruning at initialization might benefit from this scheme. Particularly, not every approximation could be used for gradual pruning as we have shown with GRASP. However, the *gradient approximation* allowed us to gradually prune while maximizing either the gradient norm or FORCE. We consider our results might encourage future work to further investigate the exploration/exploitation trade-off in pruning and find more efficient pruning schedules, not limited to the pruning at initialization.

### ACKNOWLEDGMENTS

This work was supported by the Royal Academy of Engineering under the Research Chair and Senior Research Fellowships scheme, EPSRC/MURI grant EP/N019474/1 and Five AI Limited. Pau de Jorge was fully funded by NAVER LABS Europe. Amartya Sanyal acknowledges support from The Alan Turing Institute under the Turing Doctoral Studentship grant TU/C/000023. Harkirat was supported using a Tencent studentship through the University of Oxford.

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

## A    PRUNING IMPLEMENTATION DETAILS

We present experiments on CIFAR-10/100 (Krizhevsky et al., 2009), which consists of 60k 32×32 colour images divided into 10/100 classes, and also on Imagenet challenge ILSVRC-2012 (Russakovsky et al., 2015) and its smaller version Tiny-ImageNet, which respectively consist of 1.2M/1k and 100k/200 images/classes. Networks are initialized using the Kaiming normal initialization (He et al., 2015). For CIFAR datasets, we train Resnet50[3] and VGG19[4] architectures during 350 epochs with a batch size of 128. We start with a learning rate of 0.1 and divide it by 10 at 150 and 250 epochs. As optimizer we use SGD with momentum 0.9 and weight decay $5 \times 10^{-4}$. We separate 10% of the training data for validation and report results on the test set. We perform mean and std normalization and augment the data with random crops and horizontal flips. For Tiny-Imagenet, we use the same architectures. We train during 300 epochs and divide the learning rate by 10 at 1/2 and 3/4 of the training. Other hyper-parameters remain the same. For ImageNet training, we adapt the official code[5] of Paszke et al. (2017) and we use the default settings. In this case, we use the Resnet50 and VGG19 with batch normalization architectures as implemented in Paszke et al. (2017).

In the case of FORCE and Iter SNIP, we adapt the same public implementation[6] of SNIP as Wang et al. (2020). Instead of defining an auxiliary mask to compute the saliencies, we compute the product of the weight times the gradient, which was shown to be equivalent in Lee et al. (2020). As for GRASP, we use their public code.[7] After pruning, we implement pruned connections by setting the corresponding weight to 0 and forcing the gradient to be 0. This way, a pruned weight will remain 0 during training.

An important difference between SNIP and GRASP implementations is in the way they select the mini-batch to compute the saliency. SNIP implementation simply loads a batch from the dataloader. In contrast, in GRASP implementation they keep loading batches of data until they obtain exactly 10 examples of each class, discarding redundant samples. In order to compare the methods in equal conditions, we decided to use the way SNIP collects the data since it is simpler to implement and does not require extra memory. This might cause small discrepancies between our results and the ones reported in Wang et al. (2020).

Table 2: Percentage of weights per layer for each network and dataset.

| Layer type | Conv | Fully connected | BatchNorm | Bias | Prunable | Total |
|---|---|---|---|---|---|---|
| **CIFAR10** | | | | | | |
| Resnet50 | 99.69 | 0.09 | 0.11 | 0.11 | 99.78 | 23.52M |
| VGG19 | 99.92 | 0.03 | 0.03 | 0.03 | 99.95 | 20.04M |
| **CIFAR100** | | | | | | |
| Resnet50 | 98.91 | 0.86 | 0.11 | 0.11 | 99.78 | 23.71M |
| VGG19 | 99.69 | 0.25 | 0.03 | 0.03 | 99.94 | 20.08M |
| **TinyImagenet** | | | | | | |
| Resnet50 | 98.06 | 1.71 | 0.11 | 0.11 | 99.78 | 23.91M |
| VGG19 | 99.44 | 0.51 | 0.03 | 0.03 | 99.94 | 20.13M |
| **Imagenet** | | | | | | |
| Resnet50 | 91.77 | 8.01 | 0.10 | 0.11 | 99.79 | 25.56M |
| VGG19 | 13.93 | 86.05 | 0.01 | 0.01 | 99.98 | 143.68M |

A meaningful design choice regarding SNIP and GRASP implementations is that they only prune convolutional and fully connected layers. These layers constitute the vast majority of parameters in most networks, however, as we move to high sparsity regimes, batch norm layers constitute a non-negligible amount. For CIFAR10, batch norm plus biases constitute 0.2% and 0.05% of the parameters of Resnet50 and VGG19 networks respectively. For consistency, we have as well restricted pruning to convolutional and fully connected layers and reported percentage sparsity with respect to the prunable parameters, as is also done in Lee et al. (2019) and Wang et al. (2020) to the best of our knowledge. In Table 2 we show the percentage of prunable weights for each network and

---

[3]https://github.com/kuangliu/pytorch-cifar/blob/master/models/resnet.py
[4]https://github.com/alecwangcq/GraSP/blob/master/models/base/vgg.py
[5]https://github.com/pytorch/examples/tree/master/imagenet
[6]https://github.com/mi-lad/snip
[7]https://github.com/alecwangcq/GraSP

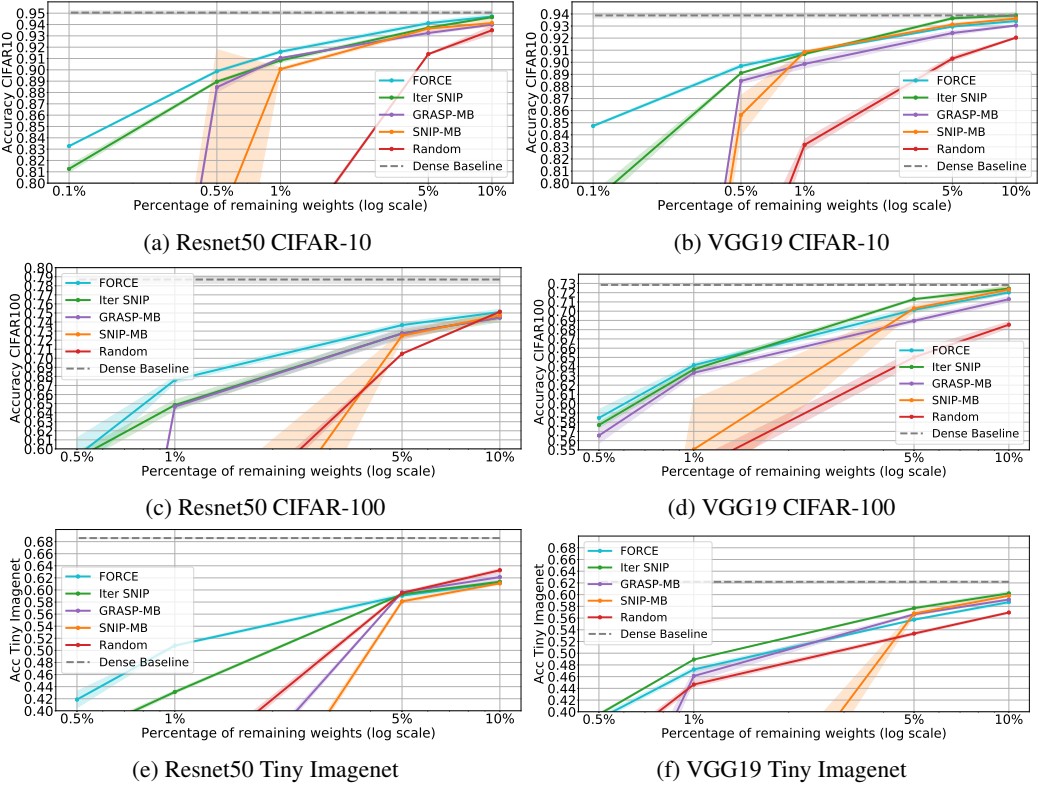

Figure 5: Test accuracies on CIFAR-10/100 and Tiny Imagenet for different pruning methods. Each point is the average over 3 runs of prune-train-test. The shaded areas denote the standard deviation of the runs (too small to be visible in some cases).

dataset we use. In future experiments we will explore the performance of pruning at initialization when including batch norm layers and biases as well.

## B    ADDITIONAL ACCURACY-SPARSITY PLOTS

In the main text we show the complete range of the accuracy-sparsity curves for the different methods so it is clear why more robust methods are needed. However, it makes it more difficult to appreciate the smaller differences at lower sparsities. In Fig 5 we show the accuracy-sparsity curves where we cut the y axis to show only the higher accuracies.

## C    FURTHER ANALYSIS OF PRUNING AT INITIALIZATION

### C.1    SALIENCY VS T

In Fig 4 (left) we have seen that for higher sparsity levels, FORCE obtains a higher saliency when we increase the number of iterations. In Fig 6 we compare the relative saliencies as we increase the number of iterations for FORCE and Iterative SNIP. As can be seen, both have a similar behaviour.

### C.2    PRUNING VS SPARSIFICATION

FORCE algorithm is able to recover pruned weights in later iterations of pruning. In order to do that, we do not consider the intermediate masks as *pruning masks* but rather as *sparsification masks*, where connections are set to 0 but not their gradients. In order to understand how does computing the FORCE (5) on a sparsified vs pruned network affect the saliency, we prune several masks with FORCE algorithm at varying sparsity levels. For each mask, we then compute their FORCE saliency either considering the pruned network (gradients of pruned connections will be set to 0 during the backward pass) or the sparsified network (we only set to 0 the connections, but let the gradient signal

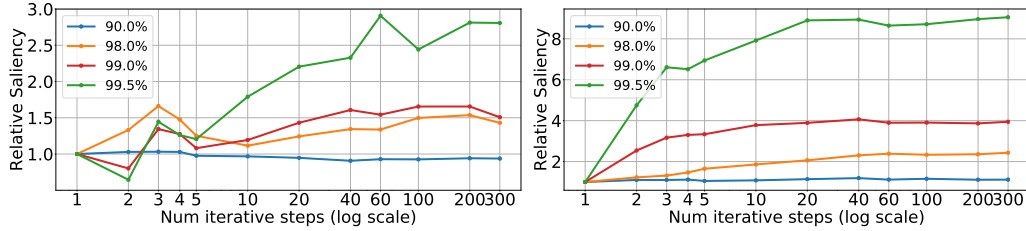

(a) Saliency vs $T$ across sparsity levels (FORCE)  (b) Saliency vs $T$ across sparsity levels (Iter SNIP)

Figure 6: FORCE saliency (6) obtained with iterative pruning normalized by the saliency obtained with one-shot SNIP, $T = 1$. (a) Applying the FORCE algorithm (b) Using Iterative SNIP. Note how both methods have similar behaviour.

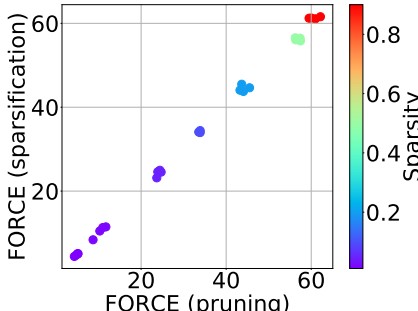

Figure 7: FORCE saliency computed for masks as we vary sparsity. FORCE (sparsification) refers to measuring FORCE when we allow the gradients of *zeroed* connections to be non-zero, while FORCE (pruning) cuts all backward signal of any *removed* connection. As can be seen on the plot, they are strongly correlated.

flow through all the connections). Results are presented in Fig 7. We observe that the two methods to compute the saliency are strongly correlated, thus, we can assume that when we use the FORCE algorithm that maximizes the saliency of sparsified networks we will also maximize the saliency of the corresponding pruned networks.

## C.3   NETWORK STRUCTURE AFTER PRUNING

In Fig 8 we visualize the structure of the networks after pruning $99.9\%$ of the parameters. We show the fraction of remaining weights and the total number of remaining weights per layer after pruning. As seen in (a) and (d), all analysed methods show a tendency to preserve the initial and final layers and to prune more heavily the deep convolutional layers, this is consistent with results reported in Wang et al. (2020). In (b) and (e), we note that FORCE has a structure that stands out compared to other methods that are more similar. This is reasonable since, it is the only method that allows pruned weights to recover. In the zoomed plots (c) and (f) we would like to point out that FORCE and Iterative SNIP preserve more weights on the deeper layers than GRASP and SNIP for VGG19 while we observe the opposite behaviour for Resnet50.

In Fig 2, we observe that gradual pruning is able to prune the Resnet50 network up to 99.99% sparsity without falling to random accuracy. In contrast, with VGG19 we observe Iterative SNIP is not able to prune more than 99.9%. In Fig 8 we observe that for Resnet50, all methods prune some layers completely. However, in the case of ResNets, even if a convolutional layer is entirely pruned, skip connections still allow the flow of forward and backward signal. On the other hand, architectures without skip connections, such as VGG, require non-empty layers to keep the flow of information. Interestingly, in (c) we observe how FORCE and Iter SNIP have a larger amount of completely pruned layers than GRASP, however, there are a few deep layers with a significantly larger amount of unpruned weights. This seems to indicate that when a high sparsity is required, it is more efficient to have fewer layers with more weights than several extremely sparse layers.

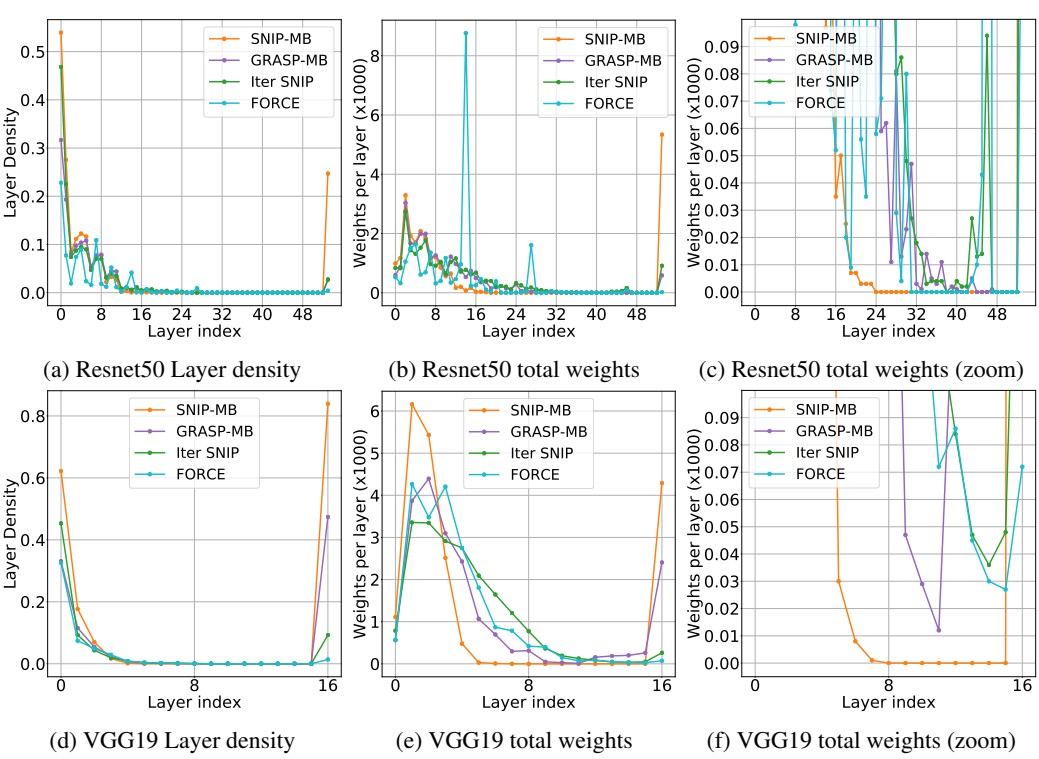

(a) Resnet50 Layer density     (b) Resnet50 total weights     (c) Resnet50 total weights (zoom)

(d) VGG19 Layer density     (e) VGG19 total weights     (f) VGG19 total weights (zoom)

Figure 8: Visualization of remaining weights after pruning 99.9% of the weights of Resnet-50 and VGG-19 for CIFAR-10. (a) and (d) show the fraction of remaining weights for each prunable layer. (b) and (e) show the actual number of remaining weights and (c) and (f) zoom to the bottom part of the plot. Observe in (c) and (f) that some layers have exactly 0 weights left, so they are removed entirely.

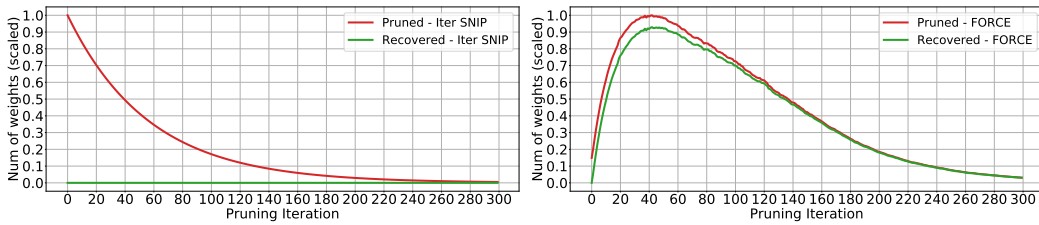

(a) Pruned and recovered weights (Iter SNIP)   (b) Pruned and recovered weights (FORCE)

Figure 9: Normalized amount of globally pruned and recovered weights at each pruning iteration for Resnet50, CIFAR-10 when pruned with (a) Iterative SNIP and (b) FORCE. As expected, the amount of recovered weights for Iterative SNIP is constantly zero, since this is by design. Moreover, the amount of pruned weights decays exponentially as expected from our pruning schedule. On the other hand, we see the amount of recovered weights is non-zero with FORCE, interestingly the amount of pruned/recovered weights does not decay monotonically but has a clear peak, indicating there is an "exploration" and a "convergence" phase during pruning.

## C.4 Evolution of Pruning Masks

As discussed in the main text, FORCE allows weights that have been pruned at earlier iterations to become non-zero again, we argue this might be beneficial compared to Iterative SNIP which will not be able to correct any possible "mistakes" made in earlier iterations. In particular, it seems to give certain advantage to prune VGG to high sparsities without breaking the flow of information (pruning a layer entirely) as can be seen in Fig 2 (b). In order to gain a better intuition of how does the amount of pruned/recovered weights ratio evolve during pruning, in Fig 9 we plot the normalized amount of pruned and recovered weights (globally on the whole network) at each iteration of FORCE and also for Iter SNIP as a sanity check. Note that Iterative SNIP does not recover weights and the amount of weights pruned at each step decays exponentially (this is expected since we derived Iterative SNIP as a constrained optimization of FORCE where each network needs to be a sub-network of the previous iteration. On the other hand, FORCE does recover weights. Moreover, the amount of pruned/recovered weights does not decay monotonically but has a clear peak, indicating there are two phases during pruning: While the amount of pruned weights increases, the algorithm explores masks which are quite far away from each other, although this might be harmful for the *gradient approximation* (refer to section 4), we argue that during the initial pruning iterations the network is still quite over-parametrized. After reaching a peak, both the pruning and recovery rapidly decay, thus the masks converge to a more constrained subset.

## C.5 Comparison with Early Pruning

For fair comparison, we provided the same amount of data to SNIP and GRASP as was used by our approach and call this variant SNIP-MB and GRASP-MB. Similarly, under this new baseline which we call *early pruning*, we train the network on 1 epoch of data of CIFAR-10, that is slightly more examples than our pruning at initialization methods which use $128 \cdot 300 = 38400$ examples (see section 5). After training for 1 epoch we perform magnitude pruning which requires no extra cost

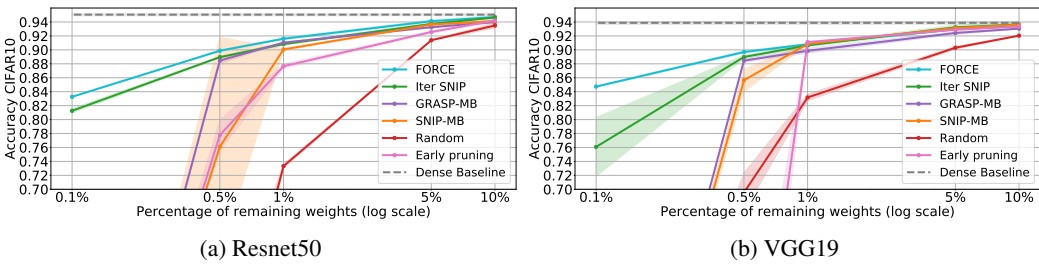

(a) Resnet50             (b) VGG19

Figure 10: Test accuracies on CIFAR-10 for different pruning methods. Each point is the average over 3 runs of prune-train-test. The shaded areas denote the standard deviation of the runs (too small to be visible in some cases). Early pruning is at most on par with gradual pruning at initialization methods and has a strong drop in performance as we go to higher sparsities.

(results presented in Fig 10). Although early pruning yields competitive results for VGG at moderate sparsity levels, it soon degrades its performance as we prune more weights. On the other hand, for Resnet architecture it is sub-optimal at all evaluated sparsity levels. Note, this result does not mean that any early pruning strategy would be sub-optimal compared to pruning at initialization, however exploring this further is out of the scope of this work.

### C.6 MOBILENET EXPERIMENTS

All our experiments were on overparameterized architectures such as Resnet and VGG. To test the wider usability of our methods, in this section we prune the Mobilenet-v2 architecture[8] (Sandler et al., 2018) which is much more "slim" than Resnet and VGG (Mobilenet has 2.3M params compared to 20.03M and 23.5M of VGG and Resnet respectively). Results are provided in Fig 11. Similarly to Resnet and VGG architectures, we see that gradual pruning tends to better preserve accuracy at higher sparsity levels than one-shot methods. Moreover, both FORCE and Iter SNIP improve over SNIP at moderate sparsity levels as well. FORCE and Iter SNIP have comparable accuracies except for high sparsity where Iter SNIP surpasses FORCE. We hypothesize for such a slim architecture ($\approx 10 \times$ fewer parameters than Resnet and VGG) the gradient approximation becomes even more sensitive to the distance between iterative masks and perhaps the exploration of FORCE is harmful in this case. As discussed in the main paper, we believe that further research to understand the exploration/exploitation trade-off when pruning might yield to even more efficient pruning schemes, especially for very high sparsity levels. We train using the same settings as described in Appendix A except for the weight decay which is set to $4 \times 10^{-5}$, following the settings of the original Mobilenet paper.

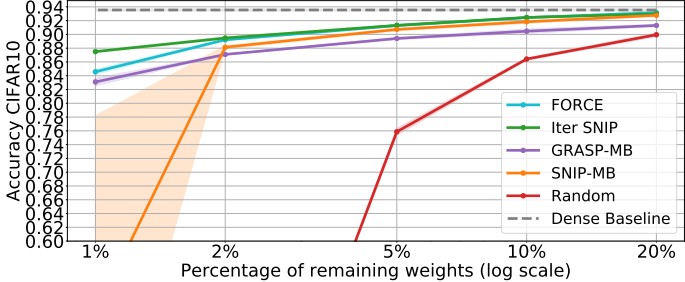

Figure 11: Test accuracies on CIFAR-10 for different pruning methods on Mobilenet architecture. Each point is the average over 3 runs of prune-train-test. The shaded areas denote the standard deviation of the runs (too small to be visible in some cases). When pruning Mobilenet-v2 architecture, which has roughly $10 \times$ less parameters than Resnet or VGG, we observe a similar pattern: Gradual pruning tends to be able to preserve accuracy for higher sparsities better than one-shot methods. Moreover, it tends to improve slightly over SNIP at moderate sparsities as well. Although GRASP does not show an acute drop in performance like SNIP, it seems to be sub-optimal with respect to other methods across all sparsities.

## D LOCAL OPTIMAL MASKS

**Definition 1** (($p, \epsilon$)-local optimal mask). Consider any two sets[9] $\mathbf{c}_t \subseteq \{1, \cdots, m\}$ and $\mathbf{c}_{t+1} \subset \mathbf{c}_t$. For any $\epsilon > 0$ and $0 \leq p \leq |\mathbf{c}_t \setminus \mathbf{c}_{t+1}|$, $\mathbf{c}_{t+1}$ is a $(p, \epsilon)$ local optimal with respect to $\mathbf{c}_t$ if the following holds

$$S\left(\theta, \mathbf{c}_{t+1}\right) \geq S\left(\theta, \left(\mathbf{c}_{t+1} \setminus S_-\right) \cup S_+\right) - \epsilon \tag{9}$$

for all $S_- \subset \mathbf{c}_{t+1}, |S_-| = p$ and $S_+ \subset \left(\mathbf{c}_t \setminus \mathbf{c}_{t+1}\right), |S_+| = p$.

**Definition 2** (CRS; Coordinate-Restricted-Smoothness). Given a function $\mathcal{L} : \mathbb{R}^m \to \mathbb{R}$ (which encodes both the network architecture and the dataset), $\mathcal{L}$ is said to be $\lambda_c$-Coordinated Restricted

---

[8]https://github.com/kuangliu/pytorch-cifar/blob/master/models/mobilenetv2.py

[9]For ease of notation, we will use this representation interchangeably with its binary encoding i.e. a $m$-dimensional binary vector with its support equal to $\mathbf{c}$

Smooth with respect to $\mathbf{c} \subseteq \{1, \cdots, m\}$ if there exists a real number $\lambda_c$ such that

$$\|\mathbf{c} \odot \nabla \mathcal{L}(\mathbf{w} \odot \mathbf{c}) - \mathbf{c} \odot \nabla \mathcal{L}(\mathbf{w} \odot \widehat{\mathbf{c}})\|_{\infty} \leq \lambda_c \|\mathbf{w} \odot \mathbf{c} - \mathbf{w} \odot \widehat{\mathbf{c}}\|_1 \tag{10}$$

for all $\mathbf{w} \in \mathbb{R}^m$ and $\widehat{\mathbf{c}} \subset \mathbf{c}$. When $s = |\mathbf{c} \setminus \widehat{\mathbf{c}}|$, an application of Holder's inequality shows

$$\lambda_c \|\mathbf{w} \odot \mathbf{c} - \mathbf{w} \odot \widehat{\mathbf{c}}\|_1 \leq \lambda_c \|\mathbf{w}\|_{\infty} \|\mathbf{c} - \widehat{\mathbf{c}}\|_1 = \lambda_c s \|\mathbf{w}\|_{\infty}$$

We define $\mathcal{L}$ to be $\Lambda$-total CRS if there exists a function $\Lambda : \{0, 1\}^m \to \mathbb{R}$ such that for all $\mathbf{c} \in \{0, 1\}^m$ $\mathcal{L}$ is $\Lambda(\mathbf{c})$-Coordinate-Restricted-Smooth with respect to $\mathbf{c}$ (for ease of notation we use $\Lambda(\mathbf{c}) = \lambda_{\mathbf{c}}$).

**Theorem 1** (Informal). *The mask $\mathbf{c}_{t+1}$ produced from $\mathbf{c}_t$ by FORCE is $\left(p, 2\lambda p \|\theta\|_{\infty}^2 |\mathbf{c}_t|\right)$-local optimal if the $\mathcal{L}$ is $\Lambda$-CRS..*

*Proof.* Consider the masks $\mathbf{c}_t$ and $\mathbf{c}_{t+1}$ where the latter is obtained by one step of FORCE on the former. Let $S_-$ and $S_+$ be any set of size $p$ such that $S_- \subset c_{t+1}$ and $S_+ \subset (c_t \setminus c_{t+1})$. Finally, for ease of notation we define $\zeta = (c_{t+1} \setminus S_-) \cup S_+$

$$
\begin{aligned}
S(\theta, \zeta) - S(\theta, \mathbf{c}_{t+1}) &= \sum_{i \in \zeta} |\theta_i \cdot \nabla \mathcal{L}(\theta \odot \zeta)_i| - \sum_{i \in \mathbf{c}_{t+1}} |\theta_i \cdot \nabla \mathcal{L}(\theta \odot \mathbf{c}_{t+1})_i| \\
&= \underbrace{\sum_{i \in \mathbf{c}_{t+1}} |\theta_i \cdot \nabla \mathcal{L}(\theta \odot \zeta)_i| - \sum_{i \in \mathbf{c}_{t+1}} |\theta_i \cdot \nabla \mathcal{L}(\theta \odot \mathbf{c}_{t+1})_i|}_{\Gamma_1} \\
&\quad + \underbrace{\sum_{i \in S_+} |\theta_i \cdot \nabla \mathcal{L}(\theta \odot \zeta)_i|}_{\Gamma_2} - \underbrace{\sum_{i \in S_-} |\theta_i \cdot \nabla \mathcal{L}(\theta \odot \zeta)_i|}_{\Gamma_3} \tag{11}
\end{aligned}
$$

Let us look at the three terms individually. We assume that $\mathcal{L}$ is $\Lambda$-CRS.

$$
\begin{aligned}
\Gamma_1 &= \sum_{i \in \mathbf{c}_{t+1}} |\theta_i \cdot \nabla \mathcal{L}(\theta \odot \zeta)_i| - |\theta_i \cdot \nabla \mathcal{L}(\theta \odot \mathbf{c}_{t+1})_i| \\
&\leq \|\mathbf{c}_{t+1} \odot \theta \odot \nabla \mathcal{L}(\theta \odot \mathbf{c}_{t+1}) - \mathbf{c}_{t+1} \odot \theta \odot \nabla \mathcal{L}(\theta \odot \zeta)\|_1 \qquad \text{By Triangle Inequality} \\
&\leq \|\mathbf{c}_{t+1} \odot \theta\|_1 \|\mathbf{c}_{t+1} \odot \nabla \mathcal{L}(\theta \odot \mathbf{c}_{t+1}) - \mathbf{c}_{t+1} \odot \nabla \mathcal{L}(\theta \odot \zeta)\|_{\infty} \qquad \text{By Holder's Inequality} \\
&\leq 2\lambda_{c_{t+1}} |c_{t+1}| p \|\theta\|_{\infty}^2 \qquad\qquad\qquad\qquad\qquad\qquad \because \mathcal{L} \text{ is } \Lambda\text{-CRS} \tag{12}
\end{aligned}
$$

$$
\begin{aligned}
\Gamma_2 &= \sum_{i \in S_+} |\theta_i \cdot \nabla \mathcal{L}(\theta \odot \zeta)_i| - |\theta_i \cdot \nabla \mathcal{L}(\theta \odot \mathbf{c}_t)_i| + |\theta_i \cdot \nabla \mathcal{L}(\theta \odot \mathbf{c}_t)_i| \\
&\leq \sum_{i \in S_+} |\theta_i \cdot \nabla \mathcal{L}(\theta \odot \mathbf{c}_t)_i| + \lambda_{\mathbf{c}_t} p \|\theta\|_{\infty}^2 (|\mathbf{c}_t| - |\mathbf{c}_{t+1}|) \qquad \because |\mathbf{c}_t \setminus \zeta| = |\mathbf{c}_t| - |\mathbf{c}_{t+1}|, \\
&\tag{13}
\end{aligned}
$$

$$
\begin{aligned}
\Gamma_3 &= -\sum_{i \in S_-} |\theta_i \cdot \nabla \mathcal{L}(\theta \odot \zeta)_i| + |\theta_i \cdot \nabla \mathcal{L}(\theta \odot \mathbf{c}_t)_i| - |\theta_i \cdot \nabla \mathcal{L}(\theta \odot \mathbf{c}_t)_i| \\
&\leq -\sum_{i \in S_-} |\theta_i \cdot \nabla \mathcal{L}(\theta \odot \mathbf{c}_t)_i| + \lambda_{\mathbf{c}_t} p \|\theta\|_{\infty}^2 (|\mathbf{c}_t| - |\mathbf{c}_{t+1}|), \tag{14}
\end{aligned}
$$

Adding eqs. (13) and (14), we get

$$
\begin{aligned}
\Gamma_2 + \Gamma_3 &\leq \sum_{i \in S_+} |\theta_i \cdot \nabla \mathcal{L}(\theta \odot \mathbf{c}_t)_i| - \sum_{i \in S_-} |\theta_i \cdot \nabla \mathcal{L}(\theta \odot \mathbf{c}_t)_i| + 2\lambda_{\mathbf{c}_t} p \|\theta\|_{\infty}^2 (|\mathbf{c}_t| - |\mathbf{c}_{t+1}|) \\
&\leq 2\lambda_{\mathbf{c}_t} p \|\theta\|_{\infty}^2 (|\mathbf{c}_t| - |\mathbf{c}_{t+1}|) - \gamma p \tag{15}
\end{aligned}
$$

where $\gamma = \min_{i \in \mathbf{c}_{t+1}, j \in (\mathbf{c}_t \setminus \mathbf{c}_{t+1})} |\theta_i \nabla \mathcal{L}(\theta \odot \mathbf{c}_t)_i| - \left|\theta_j \nabla \mathcal{L}(\theta \odot \mathbf{c}_t)_j\right| \geq 0$

Substituting eqs. (12) and (15) into (11) we get

$$S\left(\theta,\zeta\right) - S\left(\theta,\mathbf{c}_{t+1}\right) \leq 2\lambda_{\mathbf{c}_{t+1}}\left|\mathbf{c}_{t+1}\right|p\left\|\theta\right\|_{\infty}^{2} + 2\lambda_{\mathbf{c}_{t}}p\left\|\theta\right\|_{\infty}^{2}\left(\left|\mathbf{c}_{t}\right| - \left|\mathbf{c}_{t+1}\right|\right) - \gamma p$$

$$= 2\lambda_{\mathbf{c}_{t+1}}p\left\|\theta\right\|_{\infty}^{2}\left(\left|\mathbf{c}_{t+1}\right| + \left(\left|\mathbf{c}_{t}\right| - \left|\mathbf{c}_{t+1}\right|\right)\right) - \gamma p$$

$$S\left(\theta,\mathbf{c}_{t+1}\right) \geq S\left(\theta,\zeta\right) - 2\lambda_{\mathbf{c}_{t+1}}p\left\|\theta\right\|_{\infty}^{2}\left|\mathbf{c}_{t}\right| + \gamma p$$

$$S\left(\theta,\mathbf{c}_{t+1}\right) \geq S\left(\theta,\zeta\right) - 2\lambda_{\mathbf{c}_{t+1}}p\left\|\theta\right\|_{\infty}^{2}\left|\mathbf{c}_{t}\right|$$

$\square$

# E  ITERATIVE PRUNING TO MAXIMIZE THE GRADIENT NORM

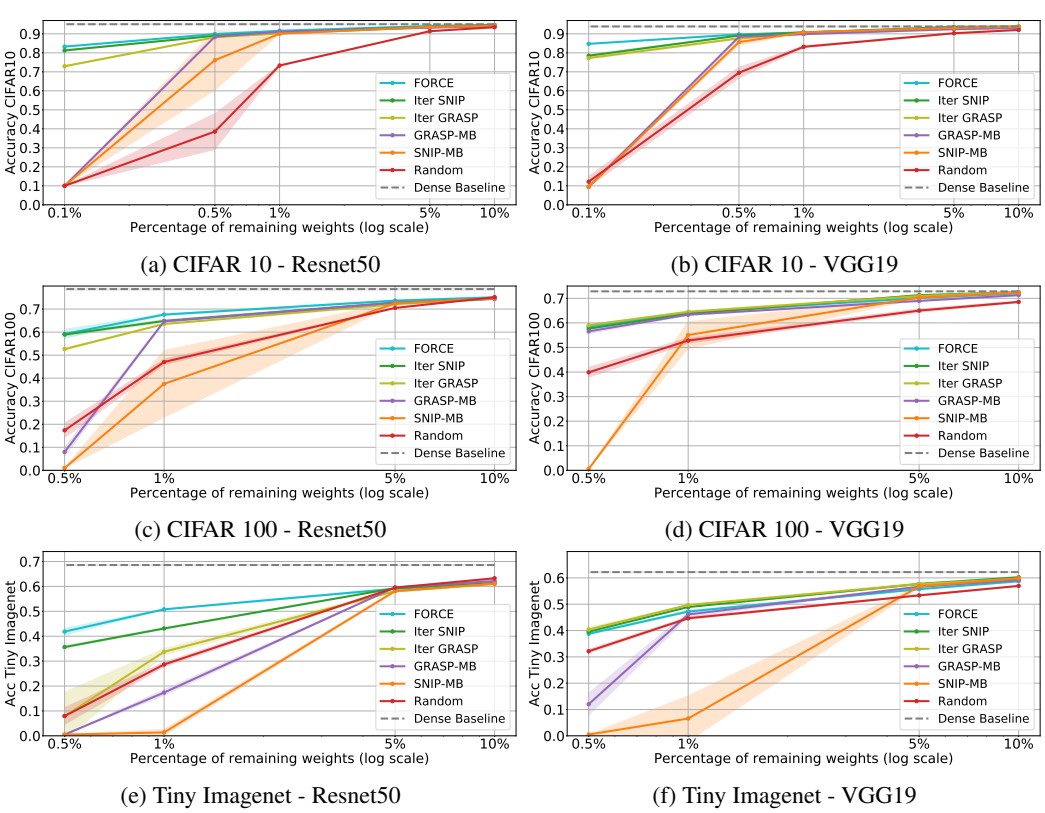

(a) CIFAR 10 - Resnet50

(b) CIFAR 10 - VGG19

(c) CIFAR 100 - Resnet50

(d) CIFAR 100 - VGG19

(e) Tiny Imagenet - Resnet50

(f) Tiny Imagenet - VGG19

Figure 12: Test accuracies for different datasets and networks when pruned with different methods. Each point is the average over 3 runs of prune-train-test. The shaded areas denote the standard deviation of the runs (sometimes too small to be visible).

## E.1  MAXIMIZING THE GRADIENT NORM USING THE GRADIENT APPROXIMATION

In order to maximize the gradient norm after pruning, the authors in Wang et al. (2020) use the first order Taylor's approximation. While this seems to be better suited than SNIP for higher levels of sparsity, it assumes that pruning is a small perturbation on the weight matrix. We argue that this approximation will not be valid as we push towards extreme sparsity values. Our *gradient approximation* (refer to section 4) can also be applied to maximize the gradient norm after pruning. In this case, we have

$$G(\boldsymbol{\theta},\boldsymbol{c}) := \Delta\mathcal{L}(\boldsymbol{\theta}\odot\boldsymbol{c}) - \Delta\mathcal{L}(\boldsymbol{\theta}) \approx \sum_{\{i:\ c_{i}=0\}} -[\nabla\mathcal{L}(\boldsymbol{\theta})_{i}]^{2}, \qquad (16)$$

where we assume pruned connections have null gradients (this is equivalent to the restriction used for Iterative SNIP) and we assume gradients remain unchanged for unpruned weights (gradient

approximation). Combining this approximation with Eq. (7), we obtain a new pruning method we name Iterative GRASP, *although it is not the same as applying GRASP iteratively*. Unlike FORCE, Iterative GRASP does not recover GRASP when $T = 1$.

In Fig 12 we compare Iterative GRASP to other pruning methods. We use the same settings as described in section 5. We observe that Iterative GRASP outperforms GRASP in the high sparsity region. Moreover, for VGG19 architecture Iterative GRASP achieves comparable performance to Iterative SNIP. Nevertheless, for Resnet50 we see that Iterative GRASP performance falls below that of FORCE and Iterative SNIP as we prune more weights. FORCE saliency takes into account both the gradient and the magnitude of the weights when computing the saliency, on the other hand, the Gradient Norm only takes gradients into account, therefore it is using less information. We hypothesize this might the reason why Iterative GRASP does not match Iterative SNIP.

### E.2 ITERATIVE GRASP (PRUNING CONSISTENCY)

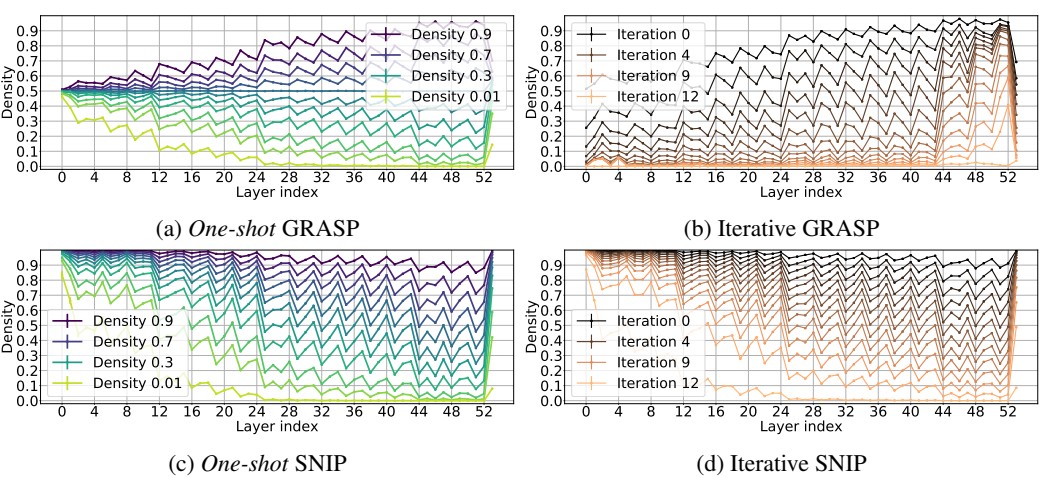

(a) *One-shot* GRASP

(b) Iterative GRASP

(c) *One-shot* SNIP

(d) Iterative SNIP

Figure 13: (a) - (c) Portion of remaining weights for each layer. Each point is an average of 3 runs and the error bars (hardly visible) denote standard deviation. (b) - (d) Portion of remaining weights for each intermediate mask that is computed during iterative pruning with global portion of remaining weights of 0.01.

As explained in the main text, we tried to apply GRASP iteratively with the Taylor approximation described in Wang et al. (2020). Unfortunately, we found that all resulting masks yield networks unable to train. In light of this result, we performed some analysis of the behaviour of GRASP compared to that of SNIP and, in the following, we provide some insights as to why we can not use GRASP's approximation iteratively.

In Liu et al. (2018), the authors show that applying a pruning method to the same architecture with different random initializations would yield consistent pruning masks. Specifically, they find that the percentage of pruned weights in each layer had very low variance. We reproduce the same experiment and additionally explore another dimension, the (global) sparsity level. Given an architecture, we prune it at varying levels of sparsity and extract the percentage of remaining weights at each layer. For each level of sparsity, we average the results over 3 trials of initialize-prune. As shown in Fig 2, both SNIP and GRASP have very low variance across initializations, on the other hand, as we vary the global sparsity with GRASP, the percentage of remaining weights for each layer is inconsistent. The layers that are most preserved at high levels of sparsity, such as the initial and last layers, are the most heavily pruned at low sparsity levels.

The authors in Liu et al. (2018), reason that manually designed networks have layers which are more redundant than others. Therefore, pruning methods even this redundancies by pruning layers with different percentages. We extend this reasoning, and hypothesize that pruning algorithms should always have preference for pruning the same (redundant) layers across all levels of sparsity. We denote this as *pruning consistency*. We observe that when applying iterative pruning to GRASP, the resulting masks tend to prune almost all of the weights at the initial and final layers, producing

networks that are unable to converge. When using iterative pruning, we prune a small portion of remaining weights at each step. Thus, we are always in the low sparsity regime, where the GRASP behaviour is reversed. Conversely, when we use SNIP the behaviour changes completely. In this case, the preserved layers are consistent across sparsity levels, and when we use iterative pruning we obtain networks that reach high accuracy values.

