# OpenReview forum: "Progressive Skeletonization: Trimming more fat from a network at initialization"
_ICLR.cc/2021/Conference — ICLR 2021 Poster_

### Official Review · AnonReviewer2 · 2020-10-22
**Solid and interesting paper that further pushes the limits of initialization pruning**

**Rating:** 6
**Confidence:** 3

**Review:**

Summary: Author observed that beyond a certain level of sparsity, existing pruning methods hat pruning at initialization performs even worse than random pruning. The author proposed FORCE and Iterative SNIP to handle this problem. These methods consistently significantly outperforms other methods on this problem.

Contributions: A new problem found in pruning. New effective methods proposed by author.

Weaknesses:
The author only shows that his methods works better than SNIP and GRASP at super high sparsity. We wonder how it performs when sparsity is not that high.
Super high sparsity problem for methods pruning at initializations is a relative small problem. The importance of this problem is weak, although author improved performance in this case. I wonder whether this is truly useful in practice.

The writing of the paper is good. It is easy to follow the paper.

It is a narrow area, but the paper is deep and through.

The Theorem 1 in the appendix seems solid, but it is acutually based on previous paper. Just a litte bit modification on the gaint's shoulder.

But overall, it is still a good paper, though the area may be very narrow.

---

> ### Author Response · Authors · 2020-11-23
> **Author Response**
>
> We thank the reviewer for the positive feedback and appreciate they find the paper well written, deep and thorough.
>
> The primary concern raised by the reviewer is regarding the usefulness of our approach in a low sparsity regime. We believe that our response below will clarify that.
>
> **Low sparsify performance**: We would like to highlight that even though the most dramatic improvements of our approaches are for high sparsities, FORCE/Iter-SNIP are at least as good as any other method in moderate sparsity regimes as well. In fact, more often than not our methods can improve upon SNIP / GRASP for moderate sparsities as well. For instance, the ResNet architecture at 90% of sparsity obtains 94.71% acc with FORCE vs 94.13% and 93.98% for SNIP and GRASP respectively. For completeness, Iter SNIP obtains 94.65% and Random pruning 93.50%. On the other hand, we would argue that the overhead of using our methods vs SNIP/GRASP is marginal compared to the cost of training, especially if we take into account the training time. Moreover, in our robustness analysis in Fig 4 (right), we show that just a few iterations can yield a large gain in performance. Thus, given the benefits of using gradual pruning and the relatively small overhead (which can be further reduced by using less iterations), we consider that FORCE or Iter SNIP should be preferred to SNIP or GRASP in any sparsity range, making our contribution more impactful.
>
> *Additionally, following AR1, we performed experiments on Mobilenet-v2 architecture on CIFAR10 and provided results in Appendix C.6. We observed that in these experiments as well, FORCE/Iter-SNIP outperformed other recent approaches.*
>
> Regarding the usefulness of high sparsity levels, we believe that high sparsity is always going to be more beneficial in terms of both time and space complexity. Also, popular deep learning frameworks like pytorch are developing torch.sparse to leverage sparse tensors (https://pytorch.org/docs/stable/sparse.html), hence we will soon be able to benefit from highly sparse neural networks.
>
> **Regarding Theorem**: Thank you for appreciating the theorem. However, we proved the theorem ourselves and are not aware of some other work having a similar proof. Would be great if you could provide the reference for the same so that we can cite the work.

---

### Official Review · AnonReviewer1 · 2020-10-26
**A good paper, nice problem fromulation**

**Rating:** 7
**Confidence:** 3

**Review:**

Summary:

This paper presented a new sensitivity-based pruning methods to cut network connections at initialisation. The proposed method FORCE shows great performance when compared to rencelty proposed SNIP and GRASP. In general, I liked the formulation of the problem, its in-depth analysis and detailed comparison to existing approaches. I see clear contribution in this paper to the field of pruning at initialisation and it is a clear accept from me.

Strength:
1. The paper comapres to very recently proposed pruning methods (SNIP and GRASP), and shows a great accuracy increase at very aggressive pruning levels.
2. The paper discussed shortcomings of previous methods in details, and clearly differentiated their proposed method in Section 4. The novelty of this paper is clear to me.
3. The paper is well written and easy to follow.
4. The paper has some interesting insights (mask zeroing, exponential scheduling, etc.) and the proposed iterative pruning at initialization is easy to apply in practice. These might add knowledge to the network pruning community.

Weakness:
1. The paper only investigates, to my understanding, ‘fat’ models. Both VGG19 and ResNet50 cannot be considered as efficient model architectures. It might be more convincing for the paper to test on recently proposed efficient network architectures such as the MobileNet family.
2. Although the paper has a potential contribution to theoretical understanding of pruning, I would like to argue aggressive pruning at initialisation cannot guarantee model’s post-training performance, and it might limit the use of this technique in reality.
Fined-grained pruning like proposed in this paper is hard to bring any real performance gains in today’s GPUs, especially when it is applied with a mask.

Despite of the weaknesses mentioned above, in general, I think this paper is well-motivated and shows a clear contribution to the community, so it is a clear accept from my side.

---

> ### Author Response · Authors · 2020-11-22
> **Author Response**
>
> We would like to thank the reviewer for such encouraging remarks and thoughtful feedback.
>
> We have added new experiments on Mobilenet and show that FORCE/Iter-SNIP outperform recent approaches in this setting as well. Details below.
>
> **Pruning fat models vs Mobilenet**: Thank you for raising this concern. All our experiments actually followed recent papers and their experiments (primarily Wang et al., 2020) so that we have the right indication of where we stand. Also, ResNet and VGG are widely used architectures. However, we do support your concern, following which we did run new experiments and pruned Mobilenet-v2 architecture on CIFAR10. We provided the results in Appendix C.6. We observe a similar overall performance pattern as with Resnet and VGG. FORCE/Iter-SNIP are the best performing methods in Mobilenet as well. We would like to thank the reviewer for suggesting this as we think this experiment helps in showing that our technique is widely applicable, and in making our experimental evaluation even more thorough.
>
> **Post-training guarantees of pruning at initialization**: We completely agree with the reviewer. There is no existing method that can provide any useful guarantees on the post-training performance of neural networks. The only way to guarantee anything would be via cross-validation, just like any other approach. We can only try to preserve some properties that we think might be useful, and verify that via experiments. Regarding computation, despite structured pruning methods having traditionally been preferred to speed-up computations, recent works have made solid advances in leveraging unstructured pruning (Elsen et al. 2020), moreover, popular deep learning frameworks like pytorch are developing torch.sparse to leverage sparse tensors (https://pytorch.org/docs/stable/sparse.html) which might make the adoption of unstructured pruning / sparsification methods much easier.
>
> References:
> Chaoqi Wang, Guodong Zhang, and Roger Grosse. Picking winning tickets before training by preserving gradient flow. In International Conference on Learning Representations, 2020. URL https://openreview.net/forum?id=SkgsACVKPH.
>
> Erich Elsen, Marat Dukhan, Trevor Gale, and Karen Simonyan. Fast sparse convnets. In Proceedings of the IEEE/CVF Conference on Computer Vision and Pattern Recognition, pp. 14629–14638, 2020.

---

### Official Review · AnonReviewer4 · 2020-10-28
**Paper that proposes iterative versions of techniques to prune at initialization, allowing for sparser networks than SNIP/GRASP**

**Rating:** 7
**Confidence:** 4

**Review:**

# Summary

The paper finds that at extreme sparsities (>95%), existing approaches to pruning neural networks at initialization devolve to worse than random pruning. The paper posits that this degenerate behavior is due to the fact that weights are pruned in groups, though the saliency metrics only capture pointwise changes. The paper presents a modified saliency metric based on SNIP, allowing for calculating salience of partially pruned networks; this in turn allows for applying an iterative version of SNIP, as well as a variant of iterative SNIP that allows for rejuvenation. These pruning techniques are evaluated, showing that they maintain accuracy at high sparsities.

# Strengths

- Well-motivated: SNIP and GRASP's drops in accuracy at high sparsities is indeed surprising and worth addressing
- Discussion of SNIP/GRASP and of Iter SNIP / FORCE is very clear
- Evaluation of methods is strong, evaluating across different target sparsities, different networks/datasets, and robustness to hyperparameter choices

# Weaknesses

- A discussion/analysis of the differences in behavior between FORCE and Iter SNIP would be warranted, since in the main body of the paper they seem fairly similar in the results. Some discussion of this is presented in App. C4, but this feels a bit lacking in conclusions. Should I always choose FORCE over Iter SNIP? Is there a reason to present Iter SNIP at all?
- The paper also needs more conceptualization and baselines for what it means to prune "at initialization". For VGG, FORCE uses $128\cdot 300 = 38,400 \approx 1 \text{epoch}$ of examples to compute the mask. Learning a mask is in fact a form of training [1]. With seeing this number of examples before computing the mask, it is plausible that a better mask could be found through other simpler means (e.g., training a network and magnitude pruning), or more complex means (e.g., [2] Figure 5 shows a VGG-19 with higher accuracy from a mask on a network that saw 6,400 examples; though [2] doesn't propose a constructive technique, it seems worth considering as an upper bound). I believe the paper would benefit from a more through discussion of what "at initialization" actually means, and what appropriate lower/upper bounds for techniques that see large numbers of examples should be.

# Overall recommendation

7: Accept

# Other comments and suggestions

- Small typo: "We" is capitalized in the middle of the first sentence of Section 5.
- Figure 4 middle: the x-axis is very hard to read. Figure 4 right: is there a difference between "batch", "b", "iter", and "it"?
- Figure 4 left: would it be possible to also include the saliency obtained from the FORCE metric in this plot?

# References used in review:

[1] Arun Mallya, Dillon Davis, Svetlana Lazebnik. "Piggyback: Adapting a Single Network to Multiple Tasks by Learning to Mask Weights".
[2] Jonathan Frankle, Gintare Karolina Dziugaite, Daniel M. Roy, Michael Carbin. "Stabilizing the Lottery Ticket Hypothesis".

---

> ### Author Response · Authors · 2020-11-22
> **Author Response**
>
> We would like to thank the reviewer for their positive and very thoughtful comments.
>
> Below we address both the weaknesses as rightly pointed out by the reviewer. We have also updated the draft accordingly (added new paragraphs and experiments in section 5.3 and appendix C.5).
>
> **Differences between FORCE and Iter SNIP**: To provide more insights on the differences between FORCE and Iter SNIP, we have added a new paragraph titled ‘FORCE vs Iter SNIP’ in section 5.3 of the main paper. Our empirical observations on a wide range of experiments suggest that even though FORCE and Iter SNIP are two conceptually different ways of optimizing the same objective, they do not perform very differently in practice. At the beginning of the pruning, FORCE, by design, supports exploration to a greater extent than Iter SNIP as it allows lost connections to get resurrected. Empirically, FORCE and Iter SNIP are consistently significantly better than the recent approaches, and FORCE is better than Iter SNIP more often than not. Thus, we would advocate using FORCE as the default choice. However, as discussed in section 5.3 as future work, a hybrid approach combining FORCE and Iter SNIP might result in superior performance.
>
> **What it means to prune at initialization**: Thank you for this question. Pruning at initialization primarily means that the training of the given downstream task (classification in our case) starts *after* the network is being pruned. Having said that, we do agree that this might not be the best definition as it means one can visit samples multiple times (pre-training) or do something more fancy/complex (e.g., self-supervised training etc.) that might require a significant amount of computation to even begin the training for the task at hand. As shown in Figure 4 in the main paper, even though the time taken by our approach is higher than SNIP/GRASP and much less than GRASP-MB, all these approaches take *negligible* amount of time compared to the total time required for the training of the downstream task. We will make this clear in the paper.
>
> For fair comparison and to make sure that it’s not just visiting the samples a priori that is giving our approach the benefit, we gave SNIP and GRASP the same advantage of using the same amount of data to find a more reliable mask, we call these approaches SNIP-MB and GRASP-MB. SNIP-MB and GRASP-MB did perform much better than SNIP and GRASP, however, FORCE/Iter-SNIP still performed significantly better than them. Hence, the optimization of our new objective function also plays a crucial role for the superior performance of FORCE/Iter-SNIP.
>
> In addition, following your question and for the sake of completeness, we added a new baseline called **early pruning** (new paragraph in section 5.3) whereby we first train the network for 1 epoch and then perform magnitude pruning. Interestingly, we find that early pruning tends to perform worse than SNIP-MB (and gradual pruning) for Resnet, and shows competitive performance at low sparsities for VGG but with a sharp drop in performance as sparsity increases. Even though these experiments again support the superiority or our approach, we would like to emphasize that they do not conclude that any early pruning strategy would be suboptimal compared to pruning at initialization as a more effective approach in this direction might require devising well thought objective function. We do believe that early pruning is of great interest, and that [2] (Stabilizing the Lottery Ticket Hypothesis) could be an “upper baseline” to pursue (not an upper bound because there is no evidence that [2] has reached the ceiling of early pruning). We leave further thoughts in this direction for future work. We again thank the reviewer for suggesting the early pruning experiment as it helped in making our analysis more complete. We will mention the suggested references in our paper.
>
>
>
> **Other comments**:
>
> - Thanks for pointing out the typo and the problem with Fig 4 x-axis.
> - There is no difference between “iter” and “it”, “batch” and “b”. We have made them all consistent.
> - Regarding Figure 4 left, we put the saliencies of both FORCE and Iter SNIP in the appendix (Figure 6). If we put both saliencies in the same figure it gets too crowded and is harder to read.

---

### Decision · Program_Chairs · 2021-01-07
**Final Decision**

**Decision:**

Accept (Poster)

**Comment:**

There is growing evidence that optimized deep networks (typically dense in the sense of nonzero parameters) often contain sparse sub-networks that can be trained from scratch to achieve similar performance as the full network. Such “skeletonization” is of obvious importance, given the rate at which deep networks in practice are increasing in size. However, many approaches to find such optimal sub-networks train the full model (and hence implicitly, the intermediate sub-networks as well), which is not a scalable path.

Some recent works show that skeletonization at initialization may provide all the efficiency benefits of sparsity, while minimally impacting accuracy. This work first notes that accuracy in such approaches degrades significantly beyond a certain level of sparsity (around 95%). One of the ideas of this work is to resurrect parameters that were pruned away earlier in this work’s iterative skeletonization, via an approach called foresight connection sensitivity (FORCE) where the “trainability”  of the pruned network is also taken into consideration. An additional idea is “Iterative SNIP”, building on the SNIP approach of Lee et al. (2019). The empirical improvements, and the observations showing the limitations of SNIP and GRASP in the regime of high sparsity, are useful.

The evaluation and overall contribution were generally appreciated by the reviewers.